

# Consequences of removal of exotic species (eucalyptus) on carbon and nitrogen cycles in the soil-plant system in a secondary tropical Atlantic forest in Brazil with a dual-isotope approach

Milena Carvalho Teixeira[1], Angela Pierre Vitória[1],
Carlos Eduardo de Rezende[1], Marcelo Gomes de Almeida[1] and
Gabriela B. Nardoto[2]

[1] Laboratório de Ciências Ambientais, Universidade Estadual do Norte Fluminense Darcy Ribeiro, Campos dos Goytacazes, Rio de Janeiro, Brazil
[2] Departamento de Ecologia, Universidade de Brasília, Brasília, Distrito Federal, Brazil

Corresponding authors
Angela Pierre Vitória,
apvitoria@uenf.br
Gabriela B. Nardoto,
gbnardoto@unb.br

## ABSTRACT

The impact of exotic species on heterogeneous native tropical forest requires the understanding on which temporal and spatial scales these processes take place. Functional tracers such as carbon ($\delta^{13}C$) and nitrogen ($\delta^{15}N$) isotopic composition in the soil-plant system might help track the alterations induced by the exotic species. Thus, we assess the effects from the removal of the exotic species eucalyptus (*Corymbia cytriodora*) in an Atlantic forest Reserve, and eucalyptus removal on the alteration of the nutrient dynamics (carbon and nitrogen). The hypotheses were: (1) the eucalyptus permanence time altered $\delta^{13}C$ and $\delta^{15}N$ in leaves, soils and litter fractions (leaves, wood, flowers + fruits, and rest); and (2) eucalyptus removal furthered decomposition process of the soil organic matter. Hence, we determined the soil granulometry, the $\delta^{13}C$ and $\delta^{15}N$ in leaves, in the superficial soil layer, and litter in three sites: a secondary forest in the Atlantic forest, and other two sites where eucalyptus had been removed in different times: 12 and 3 months ago (M12 and M3, respectively). Litter samples presented intermediate $\delta^{13}C$ and $\delta^{15}N$ values in comparison with leaves and soil. In the M3, the greater $\delta^{13}C$ values in both litter rest fraction and soil indicate the presence, cycling and soil incorporation of C, coming from the $C_4$ photosynthesis of grassy species (Poaceae). In the secondary forest, the soil $\delta^{15}N$ values were twice higher, compared with the eucalyptus removal sites, revealing the negative influence from these exotic species upon the ecosystem N dynamics. In the M12, the leaves presented higher $\delta^{13}C$ mean value and lower $\delta^{15}N$ values, compared with those from the other sites. The difference of $\delta^{13}C$ values in the litter fractions regarding the soil led to a greater fractioning of $^{13}C$ in all sites, except the flower + fruit fractions in the secondary forest, and the rest fraction in the M3 site. We conclude that the permanence of this exotic species and the eucalyptus removal have altered the C and N isotopic and elemental compositions in the soil-plant system. Our results suggest there was organic matter decomposition in all litter fractions and in all sites. However, a greater organic matter decomposition process was observed in the M3 soil, possibly because of a more intense recent input of vegetal material, as well as the presence of grassy,

easily-decomposing herbaceous species, only in this site. Therefore, the dual-isotope approach generated a more integrated picture of the impact on the ecosystem after removing eucalyptus in this secondary Atlantic forest, and could be regarded as an option for future eucalyptus removal studies.

## INTRODUCTION

Tropical forests have been reduced to fragments of their original extensions on account of activities such as urbanization, agriculture, and the replacement of native vegetation by exotic tree species for commercial purposes, as eucalyptus and pine (*Morellato & Haddad, 2000*; *Baptista & Rudel, 2006*; *Wright & Muller-Landau, 2006*; *Brockerhoff et al., 2008*; *Vitória, Alves & Santiago, 2019*). Historically in Brazil, the eucalyptus plantation has been associated with the suppression of native species, mainly in the Atlantic forest southeast region (*Brockerhoff et al., 2008*). In biome restoration-aimed efforts, this fast-growing exotic species has been used as a renewable alternative in order to connect fragments from the installation of ecological corridors, or the mixed plantation together with native species (*Brockerhoff et al., 2008*). However, the presence of eucalyptus close or consorted to secondary forests changes the natural dynamics of these forests, since this species offers fewer resources for the pollinating and dispersing fauna and which may interfere in both C and N cycles (*Guo & Sims, 1999*; *Dutta & Agrawal, 2001*). The eucalyptus litter decomposes slowly, has low nutritional attributes, and presents a higher C/N ratio in comparison to the majority of other native species from tropical forests (*Rezende, Garcia & Scotti, 2001*; *Villela et al., 2001*; *Gama-Rodrigues et al., 2008*).

In the sites destined exclusively to biodiversity protection, the eucalyptus has been suppressed from vegetation via removal, mixed plantations, and deforestation, which may cause variations in C and N cycles within the forest ecosystem compartments (*Costa, Gama-Rodrigues & Cunha, 2005*; *Gama-Rodrigues et al., 2005*). The N soil supplies are specially controlled over the climate conditions and by the vegetation (*Vitousek & Matson, 1984*). The forests stock C, which is accumulated between the photosynthesis and breathing balance in this ecosystem (*Pregitzer & Euskirchen, 2004*). Nonetheless, the C cycle and supply throughout the forest depend on its own age, from the period of natural or anthropic disturbances, and from practices of landscape usage such as deforestation and degradation (*Houghton, 2005*). Hence, the vegetation removal and/or replacement is likely to transform a C-storage ecosystem into a C-source one to the atmosphere (*Bayer et al., 2004*; *Diekow et al., 2005*).

The C and N cycles are fundamental ecological processes in the soil-plant system dynamics, thus influencing growth strategies, the subsistence of vegetal species, and the efficient applications of these elements by the community (*Vitousek & Sanford, 1986*). Within this system, the litter works as a reservoir for these and other elements. The soil organic matter decomposition rate and, consequently, the C and N release to the soil are

related to biotic factors, namely the soil microbial activity, N concentration in plants, the chemical composition of the litter material, as well as abiotic variables, as the soil humidity, temperature, texture and depth (*Stemmer, Gerzabek & Kandeler, 1998*; *Bustamante et al., 2004*; *Martinelli et al., 2009*; *Trumbore & Camargo, 2009*). The soil texture influences the dynamics of C and N, whence more clayey and humid soils present higher retention capacity of organic matter (*Stemmer, Gerzabek & Kandeler, 1998*; *Telles et al., 2003*; *Gama-Rodrigues et al., 2005*). In more sandy soils, with low retention capacity of organic matter, few differences are observed between C and N values in the soil and those of the organic litter (*Rosell, Galantini & Iglesias, 1996*). The C isotopic composition ($\delta^{13}C$) from the litter and the soil organic matter depicts the combination of past and current vegetations (*Freitas et al., 2001*; *Mardegan et al., 2009*). In the leaves, the stomas control $CO_2$ entry into the substomatal cavity, which has a direct impact on $\delta^{13}C$ present in vegetal tissues. At first, any factor that alters the atmospheric $CO_2$ isotopic ratio and/or the $c_i/c_a$ (internal C/atmospheric C) relationship, will then modify $\delta^{13}C$ (*Farquhar, O'Leary & Berry, 1982*; *Mardegan et al., 2009*; *Vitória et al., 2016, 2018*; *Teixeira et al., 2018*). Hence, in the leaves, the $\delta^{13}C$ is determined by processes which interfere and modulate photosynthesis, for example the photosynthetic syndrome, the environmental conditions (for instance irradiance, temperature, humidity), and the availability of water and nutrients (*Farquhar, Ehleringer & Hubick, 1989*; *Teixeira et al., 2015, 2018*; *Vitória et al., 2016, 2018*; *Vitória, Alves & Santiago, 2019*).

Several biogeochemical processes contribute toward the N isotopic composition ($\delta^{15}N$) in leaves, that include nitrification, mineralization, lixiviation, denitrification, volatilization, abiotic factors (temperature and humidity), bacterial colonization, time and space variations in terms of N availability in the soil, the type of N source (organic or inorganic), soil texture, among others (*Dawson et al., 2002*; *Ometto et al., 2006*; *Vallano & Sparks, 2013*). Tropical forests are dynamic environments for the N-cycle-related processes, thus resulting in N fractionation and enrichment of $^{15}N$ in the soil-plant system (*Martinelli et al., 1999*; *Nardoto et al., 2008*; *Craine et al., 2009*; *Vitória et al., 2018*). Leaf C/N ratio and soil N mineralization rates influence soil organic matter dynamics and consequently, the $\delta^{15}N$ in leaves (*Craine et al., 2009*). Soil N transformation rates and N losses to the atmosphere will, ultimately, regulate N availability in an ecosystem and consequently, the $\delta^{15}N$ in the soil-plant system (*Craine et al., 2009*). Since N availability is higher in tropical forests than in temperate forests (*Vitousek & Sanford, 1986*; *Martinelli et al., 1999*; *Boeger, Wisniewski & Reissmann, 2005*), these tropical forests will have, as a consequence, higher $\delta^{15}N$ in the soil-plant system compared to the temperate forests under lower N availability (*Craine et al., 2009*). The high amount of precipitation and almost no seasonality in temperature and humidity in those tropical forests tend to provide favorable conditions to decomposition activity by soil organisms, and consequently high decomposition rates most of the year (*Vitousek & Matson, 1984*; *Davidson et al., 2007*; *Nardoto et al., 2008*; *Viani, 2010*; *Gragnani, 2014*). Therefore, both litter decomposition and nutrient mineralization promote nutrient availability in soil, mainly N and P, thus triggering variations in $\delta^{15}N$ (*Dawson et al., 2002*; *Bustamante et al., 2004*; *Ometto et al., 2006*). Tropical and humid rainforests have higher concentrations

of C and N in the soil during rainy season (*Mazurec, 2003*; *Gama-Rodrigues et al., 2008*). The decomposition acceleration in rainy season is attributed to an increment in decomposers and macro-fauna, and, in some forests, to a more significant growth of fine roots, caused by the availability of water and nutrients (*Stemmer, Gerzabek & Kandeler, 1998*; *Bustamante et al., 2004*; *Martinelli et al., 2009*).

In the União Biological Reserve (Rebio União), a fragment of the Atlantic forest with integral nature protection located in southwest Brazil presents specific forestry management which consists on the removal of eucalyptus trees (*Corymbia citriodora*) until 2015 aiming to restore the native flora. With management, the irradiance patterns, humidity, and site temperature were altered, as well as the $\delta^{13}C$ from the main regenerating species from the understory (*Teixeira et al., 2015*, *2018*; *Vitória et al., 2016*). Other ecological processes might as well have been affected by management, like the C and N dynamics in the soil-plant system.

The objective of this work was to study the forestry management effect of eucalyptus removal in the Rebio União by determining $\delta^{13}C$ and $\delta^{15}N$ in the soil-plant system, as well as the soil granulometry in three sites: a secondary forest in the Atlantic forest, and two other sites where eucalyptus had been removed at different times: 12 months and 3 months before sampling (M12 and M3, respectively). These study hypotheses were: (1) the eucalyptus permanence time altered the $\delta^{13}C$ and $\delta^{15}N$ in leaves, soil, and organic litter fractions (leaves, wood, flowers + fruits and rest) and; (2) eucalyptus removal furthered decomposition process of the soil organic matter.

## MATERIALS AND METHODS

### Site of study, species and sampling period

This study was held in the União Biological Reserve (Rebio União), in Rio de Janeiro State, Brazil (22°27′S; 42°02′W). Field experiments were approved by SISBIO/ICMBio/MMA, to develop the project "Capacidade aclimatativa de espécies da Mata Atlântica após manejo de eucalipto na Reserva Biológica da União" at the Rebio União, Rio de Janeiro, Brazil, under the licence number 37996-3.

The climate in the region is humid tropical Aw (*Alvares et al., 2013*), with a yearly average temperature of 25 °C and an annual rainfall of 1,920 mm (85% being between October and April). This Reserve comprises around 7,800 ha of ombrophilous dense Atlantic forest, of which around 220 ha had the native tree vegetation suppressed and replaced by eucalyptus crops (*Corymbia citriodora*) (Hook.) K.D. Hill & L.A.S. Johnson. The eucalyptus tree plantations ceased to receive silvicultural treatments related to understory removal in 1996. The eucalyptus was clear-cut between 2009 and 2015.

We realized this research in three latosolic dystrophic red–yellow argisoil sites, classified according to *Empresa Brasileira de Pesquisa Agropecuária (EMBRAPA) (1999)*, *Miranda, Canellas & Nascimento (2007)* and *Dos Santos et al. (2018)*. One site is a well-preserved secondary forest; whereas the other two are eucalyptus-harvested sites, employing clear-cutting, with a single difference related to after-handling time: the M12 (where eucalyptus had been removed 12 months before our evaluation); and the M3 (where eucalyptus removal had happened three months before assessment)

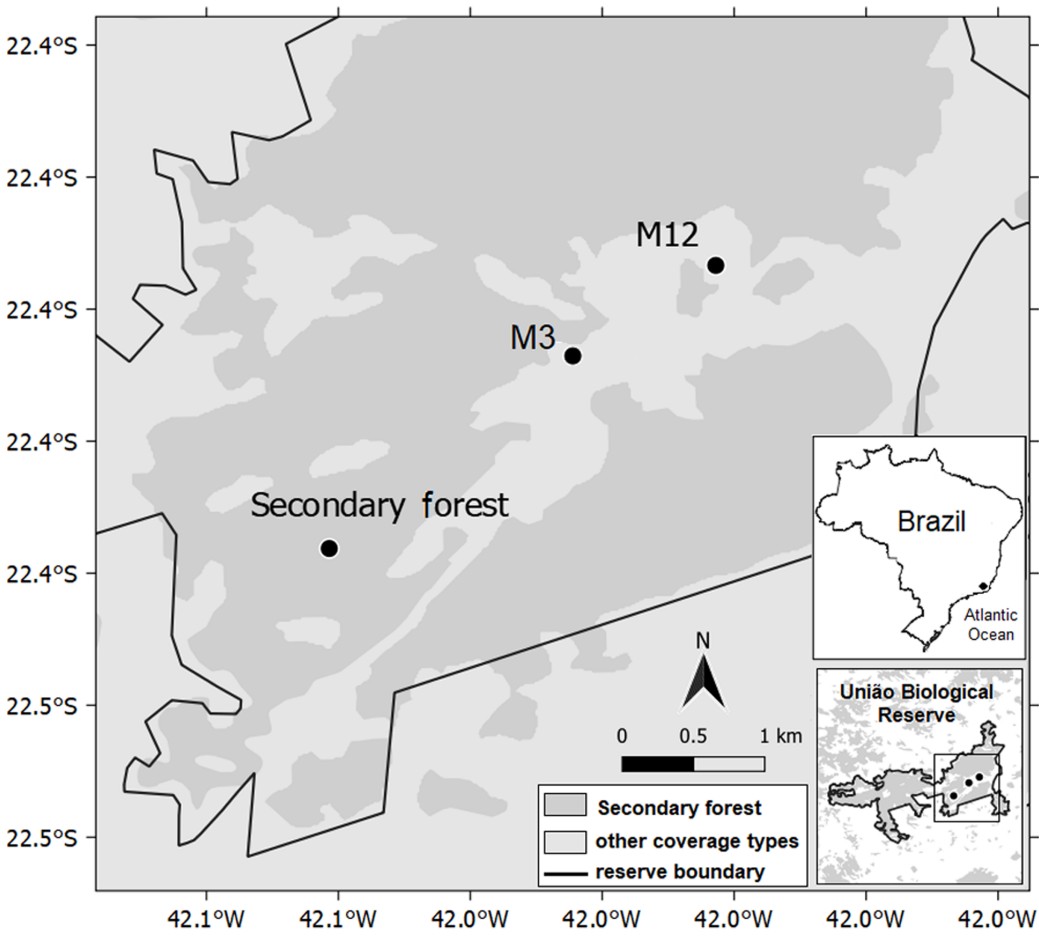

**Figure 1 Geographic map indicating the secondary forest and other two sites where eucalyptus had been removed at different times 12 and 3 months ago (M12 and M3, respectively), União Biological Reserve, Brazil.**

(Fig. 1; Table 1; Data S1). The post eucalyptus removal times were chosen according to the management activities in the Rebio União, and is sufficient to detect processes related to the decomposition of organic matter. For each of these sites, three ecosystem compartments were studied: photosynthetically active leaves, organic litter fractions, and the soil, due to its relation with organic matter decomposition processes.

For the elemental and isotopic analyses of C and N, we considered the most abundant tree species within the eucalyptus removal sites M12 and M3 (*Evaristo, Braga & Nascimento, 2011*), being all of those from the initial succession stratum (*Lorenzi, 2000*): *Siparuna guianensis* Aubl. (Siparunaceae), *Xylopia sericea* A. St.-Hill (Annonaceae), and *Byrsonima sericea* D.C. (Malpighiaceae). We collected the leaves from the eucalyptus removal sites with individuals from 2 to 5 m high, and diameter at breast height (DBH) >5 cm. For a better diversity-related sampling within the secondary forest site, we collected samples from 20 species (Data S2) according to their importance value index (IVI), all of those ranging between 10 and 15 m tall, and with a DBH >10 cm. The leaves, plant litter, and soils were collected in December 2013 during the rainy season.

Table 1 Characterization of the study sites: Secondary forest, managed eucalyptus site clear-cut 12 months ago (M12) and 3 months ago (M3) at the União Biological Reserve (Rebio União), Brazil. December 2013.—not applicable.

|  | Secondary forest | M12 | M3 |
|---|---|---|---|
| Understory of native species | Dense vegetation | Well-developed vegetation | Sparce vegetation |
| Mean irradiance ($\mu$mol m$^{-2}$ s$^{-1}$)[a] | 9 | 1.480 | 1.100 |
| Total site (ha) | – | 5.02 | 9.59 |
| Secondary forest distance (km) | – | 1.3 | 0.35 |
| Eucalyptus trees spaced (m) | – | $1.5 \times 3$[b] | $1.5 \times 3$[b] |
| Eucalyptus removal management | No management | Yes | Yes |
| Management time (months) | – | 12 | 3 |
| Presence of grasses | No | No | Yes |

Notes:
[a] Values obtained at 1:30 m in 30 points between 11:00 and 13:00 h on 15 sunny day using a quantum sensor (Li-190 coupled to Li-250 A, Li-Cor, USA).
[b] MMA-ICMBio (2008).

## Determination of the elemental and isotopic composition of C and N within the leaves, litter and soil compartments and sample number

We standardized collected samples of the third sun-exposed photosynthetically active pair of leaves, once photosynthetic activity and irradiance variation are important factors to determine $\delta^{13}$C (Vitória et al., 2016). Young leaves are not photosynthetically active and enriched in $^{13}$C (Cernusak et al., 2009), similar to shadow leaves (Vitória et al., 2016). Besides, great variation can be found in $\delta^{13}$C within a single tree crown on account of the variations in environmental conditions (Le Roux et al., 2001; McDowell et al., 2011). Thus, sun-exposed branches of individuals of the secondary forest were collected from the canopy by using high pruning shears. Sun-leaves from eucalyptus removal areas (M3 and M12 sites) were manually collected. For each species, 5–7 leaves per individual were sampled and then dried in an oven at 50 °C for 48 h (Vitória et al., 2016). Afterward the leaves were macerated in liquid N and homogenized per individual. Regarding the eucalyptus removal sites, the samples collected comprise 5–15 individuals ($n = 5$–15) of the three species from the regenerating understory in both the M12 and the M3 sites. For a better diversity-related sampling within the secondary forest site, we collected samples from 20 species (Data S2). For all of these 20 species, 5 adult individuals had their leaves sampled, except for the following species: Ficus gomelleira ($n = 4$), Cupania racemosa ($n = 4$), Brosimum glazioui ($n = 4$), Virola gardneri ($n = 2$), Micropholis guianensis ($n = 2$), Ocotea diospyrifolia ($n = 2$). Thus, for the secondary forest, we collected samples from 2 to 5 individuals ($n = 2$–5).

The litter layer on the soil was collected via a $20 \times 20$ cm metal frame randomly arranged 25 times in each of the three study sites, totaling 75 samples with 400 cm$^2$ of litter. The collect points were chosen at random. The collected material was dried at constant weight in an oven at 50 °C for 48 h (Mayor et al., 2014; Camara et al., 2018). These samples were screen-sorted by the following fractions: leaves (leaves and leaflets, whole or fragmented); wood (branch or bark pieces with a diameter smaller than 2 cm, and

fragments), flower + fruits (flowers, inflorescence or organ, and fragments recognized as such, fruits and seeds, whole or fragmented), and the rest (residues smaller than 2 mm, animal-origin and unknown matter) (*Proctor et al., 1983*). The 25 dried litter fractions were macerated and grouped in 5 composed samples for each soil-litter fraction per site ($n = 5$).

The superficial soil (0–10 cm) was collected from below the litter layer withdrawn, totalizing 75 samples with 400 cm$^2$ of soil. The 75 soil samples were dried at constant weight in an oven at 60 °C for 48 h (*Mayor et al., 2014*; *Severo et al., 2017*, *Camara et al., 2018*). No decarbonation was performed once previous tests verified the absence of inorganic carbon in the soil (*Camara et al., 2018*). A soil sub-sample was used to determine granulometric fractions (item 3), and another sub-sample was sieved in a 2 mm mesh and homogenized for isotopic and elemental determination of C and N ($n = 25$) (*Telles et al., 2003*).

Leaves, organic litter fractions, and soils were dried, ground, weighed (1 mg approximately), and inserted in tin capsules for C and N elemental and isotopic determination. All samples were combusted in an elemental analyzer (Flash 2000 Organic Elemental Analyser) which measured elemental concentrations of C and N; coupled to a stable isotope ratio mass spectrometer (IR-MS Delta V Advantage; Thermo Scientific, Waltham, MA, USA); and combined with a ConFloIV Interface (Thermo Scientific, Waltham, MA, USA), which measured the isotopic composition of C and N. Pee Dee Belemnite (PDB) and atmospheric N were used as standard values for C and N analyses, respectively. The values of PDB $\delta^{13}$C and atmospheric $\delta^{15}$N are respectively, 0.0111802‰ and 0.0036‰. The analytical precision was ±0.1 for $\delta^{13}$C and ±0.2 for $\delta^{15}$N, and the accuracy for elemental and isotopic compositions were determined by certified Organic Analytical Standard (OAS): Wheat Flour Standard OAS/Isotope Cert. 114858 and Low Organic Content Soil Standard OAS/Isotope Cert. 201613. The values of OAS 114858 were 39.38 ± 0.19% for C; 1.36 ± 0.23% for N; −27.21 ± 0.13‰ for $\delta^{13}$C; 2.85 ± 0.17‰ for $\delta^{15}$N. The values of OAS 201613 were 1.52 ± 0.02% for C; 0.13 ± 0.02% for N; −27.46 ± 0.11‰ for $\delta^{13}$C; 6.70 ± 0.15‰ for $\delta^{15}$N.

Carbon and nitrogen stable isotope values are reported in "delta" notation as $\delta$ values in parts per thousand (‰).

## Determination of the soil granulometric fractions

After soil preparation procedures, as aforementioned, sub-samples were sieved, separating >2 mm fractions by sifting in successive intervals. Soil texture (granulometry) was determined in the soil fraction <2 mm. The methodology used here followed *Wentworth (1922)* by using a laser diffraction particle analyzer (SALD-3101; Shimadzu, Kyoto, Japan).

For precision analytical control, we measured the analytical variation among the analytical triplicates in every 20 samples, with an acceptable variation coefficient inferior to 10%. The accuracy surpassed 90%. Three certified samples supplied by the manufacturer of the equipment (Shimadzu, Kyoto, Japan) with differentiated particle size range (JIS class 11, Licopodium and glass beads, Data S3) were used to evaluate the accuracy of the soil granulometric fractions. The limit of detection herein was 0.1%.

The granulometric distribution was compiled by the software SysGran (3.0), and the data was grouped in sand, silt, and clay. The data are expressed in mean and standard error values.

## Statistical analysis

In order to assess the normality in the data, the Shapiro Wilk ($P \leq 5\%$) test was conducted. After assessing the normality and homogeneity, the data underwent parametrical analyses. We analyzed differences between means via ANOVA (two-way), followed by Tukey test ($P \leq 5\%$). The factors considered herein were the compartments (leaves, litter fractions, and soil), sites, and granulometric fractions; it all aims to understand the interference of eucalyptus permanence time in the $\delta^{13}$C and $\delta^{15}$N in leaves, soil and organic litter fractions. The variance analysis was realized in the software Statistica 7.0. We calculated the sand regression coefficient correlations with Sigma Plot 11.0 software package Statistical Package for the Social Sciences (SPSS).

The C isotopic fractionation during organic matter decomposition was calculated from the difference between the $\delta^{13}$C in the soil and the litter fractions. The same was employed to N, by using $\delta^{15}$N. We inferred the soil organic matter decomposition based on these calculations.

## RESULTS

### The relationship between C and N in the soil-plant system

The values of $\delta^{13}$C and $\delta^{15}$N were higher in the soil > litter > leaves for all studied sites (Fig. 2; Data S4–S6). We observed a direct relationship between $\delta^{13}$C and $\delta^{15}$N (Figs. 2A, 2C and 2E), and an inverse one between $\delta^{15}$N and N in all three sites (Figs. 2B, 2D and 2F).

Regardless of the site, the elemental concentration values for C and for the C/N ratio were respectively distributed: litter fractions > leaves > soil (Table 2). For N, leaves had higher values than litter, being its concentration thus distributed: leaves > litter fractions > soil (Table 2). Significant higher concentrations of C and N in the soil were found in the M12 site, when compared with the secondary forest and to the M3 site (Table 2). The N concentrations of all plant litter fractions were significantly smaller in the two eucalyptus removal sites (M12 and M3, Table 2). The C/N ratio in the leaves was higher in the M12 site. Higher C/N ratio values in wood fractions were obtained in the eucalyptus removal sites, in comparison with those of the secondary forest (Table 2).

### Decomposition of organic matter in the litter fractions

There was an enrichment in the $^{13}$C between litter fractions and soil in all sites (except for flower and fruit fractions from the secondary forest and the fraction of the rest in the M3 site) (Table 3). On the other hand, there was no enrichment in $^{15}$N in any of the studied sites, being observed only negative values at the time of calculating differences between litter fractions to the soil, for each site (Table 3).

There was a negative relationship between $\delta^{15}$N and the C/N ratio in the soil (Fig. 3), having the secondary forest the highest value for $\delta^{15}$N in comparison to the eucalyptus

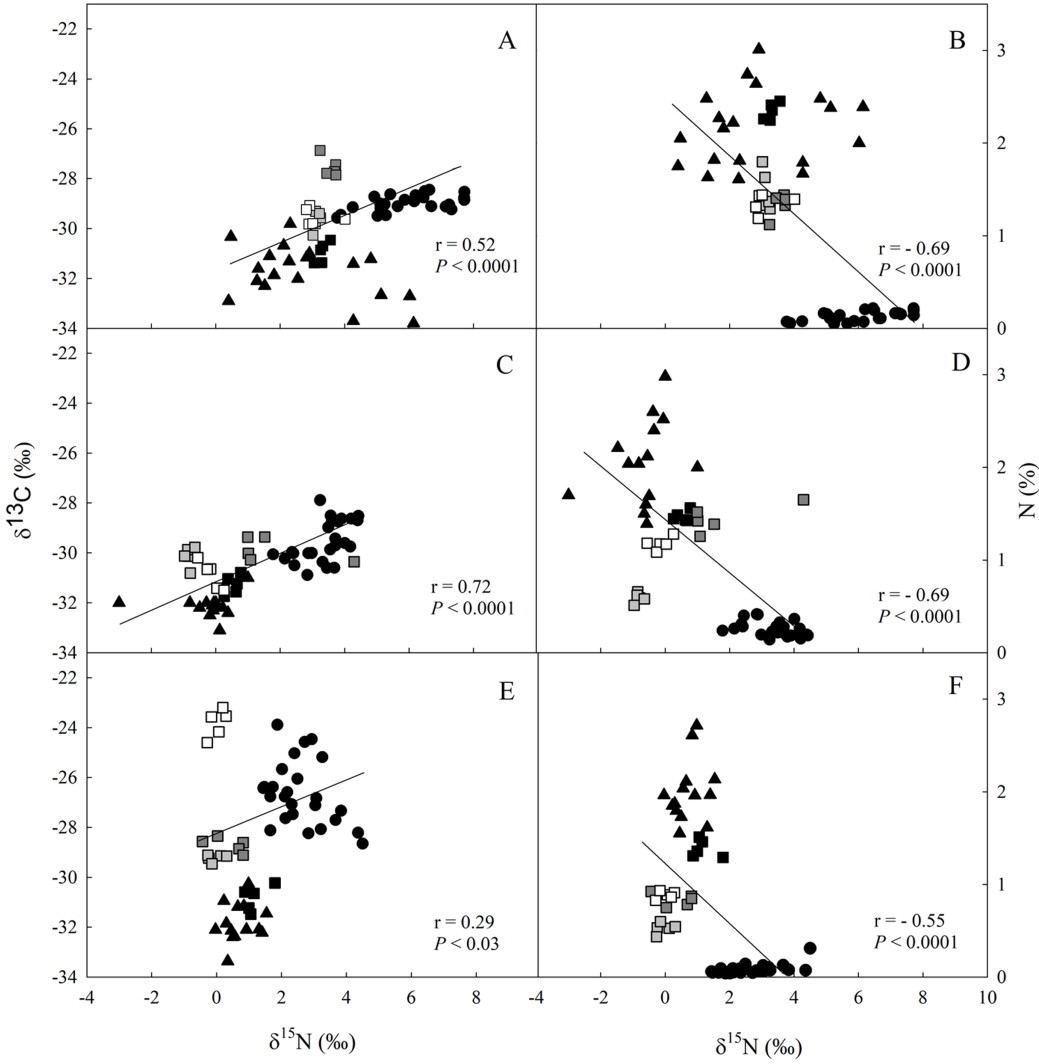

**Figure 2 Linear regression between nitrogen (δ¹⁵N) and carbon (δ¹³C) isotopic composition and between δ¹³C and the elemental concentration of N (%) in different ecosystem compartiments for studied sites at the União.** Linear regression between nitrogen (δ¹⁵N) and carbon (δ¹³C) isotopic composition (A, C and E), and between δ¹³C and the elemental concentration of N (%) (B, D and F) for leaves (closed circle), litter fractions (squares: white—rest fraction; dark grey—fruits + flowers fraction; light grey—wood fraction; black—leaf fraction) and soil (closed triangle) in the secondary forest (A and B); site managed 12 months ago (M12) (C and D); and site managed 3 months ago (M3) (E and F) in União Biological Reserve (Rebio União), Brazil. December, 2013.

removal sites, which did not vary between one another. The soil C/N ratio was higher in the most recent eucalyptus removal site (M3) than in the M12 site and the secondary forest (Fig. 3).

The secondary forest soil, where greater values of δ¹⁵N were found, presented higher clay content, whereas in the M3 site, which presented a higher C/N ratio, had also featured a more significant amount of sand (Fig. 4; Data S7).

**Table 2 Mean and standard error (SE) of the total concentration for C, N, and the C/N ratio in leaves, litter fractions and soil in the studied sites at the União Biological Reserve.**

| Site | Fraction | C (%) | N (%) | C/N |
|---|---|---|---|---|
| Secondary forest | Leave | 42.4 ± 0.4 Bb | 2.1 ± 0.1 Aa | 21.1 ± 0.5 Bb |
| | Leaves litter | 46.9 ± 0.7 Ba | 2.3 ± 0.1 Aa | 23.4 ± 0.5 Cb |
| | Residue litter | 50.1 ± 0.6 Aa | 1.4 ± 0.1 Ab | 43.4 ± 1.3 Ba |
| | Flower + fruit litter | 47.7 ± 0.1 Aa | 1.3 ± 0.1 Ab | 42.0 ± 1.9 Ba |
| | Wood litter | 46.5 ± 1.9 Aa | 1.5 ± 0.1 Ab | 36.9 ± 1.4 Ba |
| | Soil | 1.5 ± 0.1 Bc | 0.1 ± 0.1 Bc | 16.5 ± 2.8 Ac |
| M12 | Leave | 44.8 ± 0.7 Ab | 2.6 ± 0.2 Aa | 44.8 ± 2.4 Ab |
| | Leave litter | 48.6 ± 0.4 Ba | 1.5 ± 0.1 Ba | 38.6 ± 0.5 Bc |
| | Residue litter | 48.4 ± 0.6 Aa | 1.2 ± 0.1 Bb | 48.0 ± 0.8 Bb |
| | Flower + fruit litter | 49.8 ± 0.3 Aa | 1.5 ± 0.1 Aa | 40.5 ± 1.6 Bb |
| | Wood litter | 48.4 ± 0.8 Aa | 0.6 ± 0.1 Bc | 96.3 ± 4.5 Aa |
| | Soil | 3.3 ± 0.1 Ac | 0.3 ± 0.1 Ad | 15.8 ± 0.8 Ad |
| M3 | Leave | 45.2 ± 0.5 Aa | 2.0 ± 0.2 Ba | 23.4 ± 0.9 Bc |
| | Leaves litter | 51.4 ± 1.1 Aa | 1.4 ± 0.1 Ba | 43.3 ± 0.9 Ab |
| | Residue litter | 45.1 ± 0.4 Ba | 0.9 ± 0.1 Cb | 59.6 ± 1.4 Aa |
| | Flower + fruit litter | 46.3 ± 0.6 Aa | 0.8 ± 0.1 Bb | 65.1 ± 2.7 Aa |
| | Wood litter | 47.5 ± 0.6 Aa | 0.5 ± 0.1 Bb | 106.2 ± 5.9 Aa |
| | Soil | 1.5 ± 0.1 Bb | 0.2 ± 0.1 Bc | 21.2 ± 0.7 Ac |

Note:
Mean and standard error (SE) of the total concentration for C, N, and the C/N ratio in leaves, litter fractions (leaves, residue, flower + fruits, and wood) and soil in the secondary forest, sites managed 12 months ago (M12), and sites managed 3 months ago (M3) at the União Biological Reserve (Rebio União), Brazil. December 2013. Uppercase letters compare the sites and the lowercase letters compare the compartments.

**Table 3 Organic matter decomposition from the difference between $\delta^{13}C$ and $\delta^{15}N$ values of the soil and the different litter fractions for the secondary forests.**

| Site | Decomposition | $\delta^{13}C$ (‰) | $\delta^{15}N$ (‰) |
|---|---|---|---|
| Secondary forest | Leave litter | 2.0 | −2.7 |
| | Residue litter | 0.5 | −2.9 |
| | Flower + fruit litter | −1.5 | −2.4 |
| | Wood litter | 0.7 | −2.9 |
| M12 | Leave litter | 1.7 | −2.8 |
| | Residue litter | 1.3 | −3.3 |
| | Flower + fruit litter | 0.3 | −1.6 |
| | Wood litter | 0.6 | −2.5 |
| M3 | Leave litter | 4.2 | −1.4 |
| | Residue litter | −2.8 | −2.6 |
| | Flower + fruit litter | 2.0 | −2.2 |
| | Wood litter | 2.6 | −2.6 |

Note:
Organic matter decomposition from the difference between $\delta^{13}C$ and $\delta^{15}N$ values of the soil and the different litter fractions (leaves, residue, flower + fruits, and wood) at the União Biological Reserve (Rebio União), Brazil in secondary forest, site managed 12 months ago (M12) and site managed 3 months ago (M3).

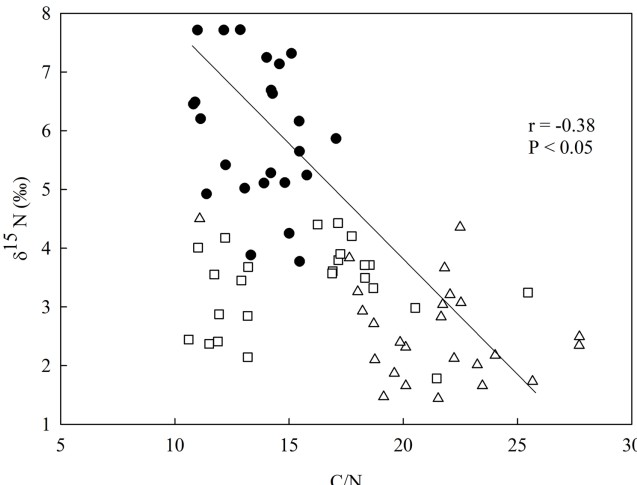

**Figure 3 Linear regression between nitrogen isotopic composition ($\delta^{15}$N) and the in-soil C/N ratio in the studied sites at the União Biological Reserve (Rebio União), Brazil.** Linear regression between nitrogen isotopic composition ($\delta^{15}$N) and the in-soil C/N ratio in the secondary forest (black circles), the site managed 12 months ago (M12, white square), and the site managed 3 months ago (M3, grey triangle) at the União Biological Reserve (Rebio União), Brazil. December, 2013.

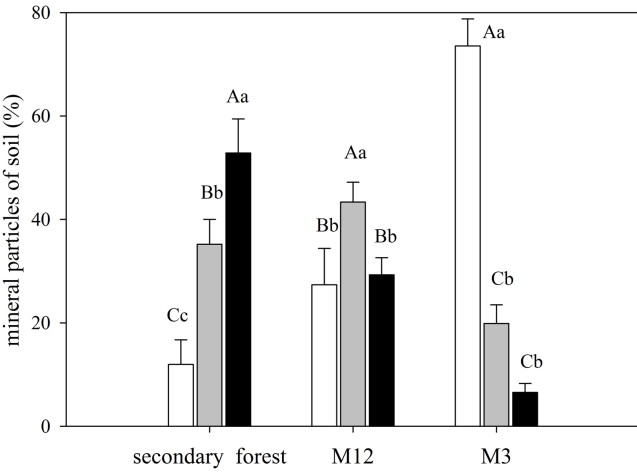

**Figure 4 Mean and standard error (SE) of sand, silt, and clay percentage for the studied sites at the União Biological Reserve.** Mean and standard error (SE) of sand percentage (2 mm–63 μm) in white; silt (63 μm–4 μm) in light grey; and clay (<4 μm) in black in the superficial soil layer in secondary forest; site managed 12 months ago (M12); and site managed 3 months ago (M3) in União Biological Reserve (Rebio União), Brazil. December, 2013. Uppercase letters compare between sites, and lowercase letters compare between fractions.

## Comparison of $\delta^{13}$C and $\delta^{15}$N between the compartments and sites

The $\delta^{13}$C values of soil varied from −26.7‰ and −29.6‰ and were consistently higher than those found in leaves and leaf litter fractions (Fig. 5A). The mean soil $\delta^{13}$C in the M3 site was higher than those from both the secondary forest and the M12 site. Significant differences were observed in $\delta^{13}$C values of leaves for the M12 site, in comparison to the other two sites; and also with foliar $\delta^{13}$C values ranging from −31.6‰ (secondary

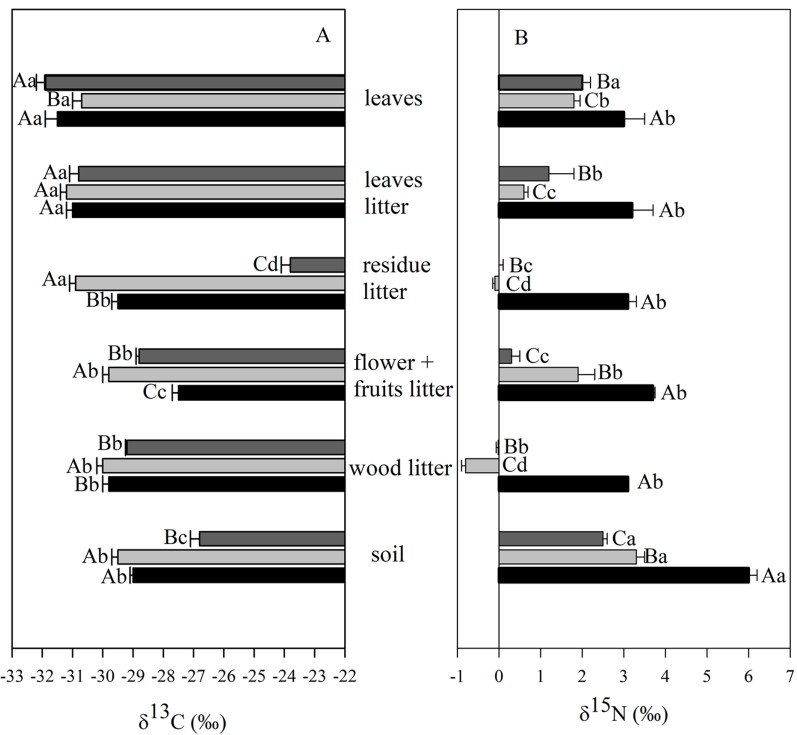

**Figure 5 Mean and SE (standard error) of carbon (δ¹³C) and nitrogen (δ¹⁵N) isotopic composition in different ecosystem compartments at the União Biological Reserve.** Mean and SE (standard error) of carbon ($\delta^{13}C$) (A) and nitrogen ($\delta^{15}N$) (B) isotopic composition in leaves, litter fractions (leaves, residue, flower + fruits and wood) and soil from secondary forest (black); site managed 12 months ago (M12) (light grey); and site managed 3 months ago (M3) (dark grey) in União Biological Reserve (ReBio União), Brazil. December, 2013. Uppercase letters compare between sites, and lowercase letters compare between compartments.

forest) to −32.2‰ (M3 site). The $\delta^{13}C$ value of the litter rest fraction in the M3 site was higher than those from the secondary forest and the M12 site (Fig. 5A). In the organic litter, we found that the rest fraction presented the highest $\delta^{13}C$ value (−23.8‰, M3 site), whereas the lowest value was found in the leaf fraction (−31.3‰, M12 site).

The $\delta^{15}N$ values have significantly been higher in the secondary forest samples, when compared with those from the eucalyptus removal sites, regardless of the compartment under analysis (Fig. 5B). Concerning the soil, only positive values were verified, which varied from 2.6‰ to 6.0‰. Positive values were also observed for leaves, going from 1.6‰ to 2.9‰. Negative values of $\delta^{15}N$ were registered only in a few litter fractions (rest and wood) from the M12 site.

## DISCUSSION

### Variation of carbon isotopic composition (δ¹³C) in all compartments in the eucalyptus removal sites

The different $\delta^{13}C$ variations of compartments between the eucalyptus removal sites and the secondary forest are mainly caused by the difference in the floristic composition, the organic litter input, and the irradiance, which consequently promotes variations in
temperature and humidity, thus affecting processes such as assimilation of C and decomposition. The eucalyptus removal has altered the $\delta^{13}$C in every compartment of the managed sites, in comparison with the secondary forest. The higher $\delta^{13}$C values of the residue litter fraction and the soil in the M3 site reflect the presence, and carbon soil incorporation originated from the photosynthesis of Poaceae species (grassy), with photosynthetic $C_4$ syndrome (*Farquhar, 1983*). The regenerating understory of tree species in the Atlantic forest in the M3 site was less developed than in the M12 site, possibly on account of the less time for native vegetation to grow and the competition with grassy that is abundant in the M3 site. $C_4$ species present a greater $\delta^{13}$C (ranging from −15‰ to −9‰) than that of the tree species, which feature photosynthetic $C_3$ syndrome (whose $\delta^{13}$C values may vary from −35‰ to −23‰) (*Farquhar, Ehleringer & Hubick, 1989*; *Martinelli et al., 2009*). The grassy species contributed to the organic matter onto the soil, raising the $\delta^{13}$C of soil from the M3 site. In the secondary forest, $\delta^{13}$C values were higher in the sequence soil > litter fractions > leaves. A similar pattern was observed in the eucalyptus removal sites, except for the rest litter fraction in the M3 site given the reason aforementioned. The highest soil $\delta^{13}$C value, in comparison with the remaining analyzed compartments, may be explained by the fast mineralization of the components with lower $\delta^{13}$C values (*Gerschlauer et al., 2018*). Differences between the $\delta^{13}$C from the distinct litter fractions and the $\delta^{13}$C from the soil are interpreted as consequences of the litter organic matter decomposition (*Gerschlauer et al., 2018*). During decomposition of litter, fractionation occurs due to the decomposers that preferentially use $^{12}$C resulting in $^{13}$C-enrichment in the remaining soil organic matter (*Lichtfouse et al., 1995*). The difference in $\delta^{13}$C value ($\delta^{13}$C litter − $\delta^{13}$C soil) and the low C/N ratio in the soil, observed in the M3 site, agree with the idea about altering the dynamics of this soil organic matter.

Our results suggest there was organic matter decomposition in every litter fraction and in every site. However, a higher decomposition of the organic matter determined from the $\delta^{13}$C in the M3 site was also observed. This result possibly comes from the more significant recent input of vegetal matter into the M3 site soil, as well as the presence of grassy (herbaceous species of rapid decomposition) exclusively in this site. Despite the soil humidity reduction caused by microclimatic alterations from recent handling in the M3 site (such as less humidity, higher irradiance, temperature, and winds), the input increase was enough to evidence more significant decomposition in this particular site.

The highest $\delta^{13}$C values found in the M12 site leaves, compared to both secondary forest and the M3 sites, are explained by the fact that this vegetation was formed under greater irradiance, for 1 year after handling. Sites exposed to more intense irradiance provide the environmental conditions which increase the $\delta^{13}$C values within the photosynthesizing tissues (*Ehleringer et al., 1986*; *Domingues et al., 2005*; *Van der Sleen et al., 2014*; *Teixeira et al., 2015*; *Vitória et al., 2016*).

In all sites, the $\delta^{13}$C values from the litter leaf fractions were a reflex of the $\delta^{13}$C living leaves. However, despite the previous discussion, the influence of the irradiance variation upon $\delta^{13}$C values in leaves was not the single cause for $\delta^{13}$C values in the litter rest fraction, mainly in the M3 site, where grassy ($C_4$ species) enrich the soil with $^{13}$C.

Compared with litter leaves, the remaining fractions ("flowers + fruits" and wood) presented higher $\delta^{13}C$ values. Variations between the $\delta^{13}C$ from the source and sink organs have been widely reported, is the enrichment extension variation of the sink organs $\delta^{13}C$ dependent on the tissue, species, and environmental conditions (*Cernusak et al., 2009*; *Vitória et al., 2016*).

Among the causes of isotopic variation between source and sink is the variability in the biochemical tissue composition (*Cernusak et al., 2009*). The leaves present higher cellulose concentrations than do stems and roots which possess more lignin. After eucalyptus removal in the M3 and the M12 sites, the cellulose from leaves decomposed faster than did the lignin from stems, thus changing the soil $\delta^{13}C$. Cellulose is more enriched with $^{13}C$ than lignin and lipids (*Badeck et al., 2005*; *Cernusak et al., 2009*). Another isotopic difference-contributing factor between source and sink is the isotopic fractionation during night breathing. The $CO_2$ respired in the dark by the aerial part is significantly enriched with $^{13}C$ when compared with the respiratory substrate pool, whereas the $CO_2$ aspired by the roots does not present enrichment (*Klumpp et al., 2005*). Moreover, stems grow preferentially at night (*Steppe et al., 2005*; *Saveyn, Steppe & Lemeur, 2007*). This is when the exported carbohydrates present higher $\delta^{13}C$ values. As a consequence, the wood features enrichment of $^{13}C$ when compared with the leaves. Also, carbohydrates exported during the day present lower $\delta^{13}C$ values, and supply preferentially for the leaf growth, which occurs faster during the day (*Walter & Schurr, 2005*).

However, it is essential to consider that the nature of the site sampling, which occurred under a unique field experimental condition, prevents the replication so as to increase the robustness of this data. In this way, although the data indicate that the presence of grassy and the higher irradiance was determining factors for the alteration of $\delta^{13}C$ in the analyzed compartments, these data should be regarded cautiously.

## Differences in nitrogen isotopic composition ($\delta^{15}N$) in the soil-plant system related to eucalyptus removal

The higher $\delta^{15}N$ values within the soil-plant system in the secondary forest, compared to the eucalyptus removal sites, can relate to differences in the composition of vegetal species, and in the strategies employed by them regarding the use of the soil nutrients (*Viani, 2010*). Depending on the variation in the functional diversity among coexisting species, one expects a more efficient use of resources and, in parallel, a reduction in the interspecific competition which, consequently, may increase the ecosystem productivity (*Naeem et al., 2009*). Some species may also disproportionately contribute to the ecosystem-related processes on account of a high specialization level, as the symbiotic nitrogen fixation (*Reed, Cleveland & Townsend, 2008*). Although tropical forests have an immense nutrient stock, forest sites under management, in general, may lack nutrients, thus presenting a decrease in these stocks (*Bellote, Dedecek & Silva, 2008*). Besides, other factors such as the dominant species, the management period and activity type, the rain cycles and intensity, and the soil properties interfere in the dynamics of nutrients (*Bellote, Dedecek & Silva, 2008*).

Our data shows the influence of eucalyptus removal on the superficial soil layer, being the eucalyptus removal sites (M3 and M12), the ones to present lower $\delta^{15}N$ in comparison

with the secondary forest. Similar $\delta^{15}N$ values, to those from the secondary forest, were also found in: other Atlantic forest soils (between −1.2‰ and 6.0‰) (*Viani, 2010*; *Gragnani, 2014*); in the litter (between −1.8‰ and 1.8‰) (*Owen, 2013*); and in tree leaves from tropical forests (between −3.0‰ and 12.0‰) (*Davidson et al., 2007*; *Nardoto et al., 2008*, *Vitória et al., 2018*).

Other aspects regarding the retention of N and the difference in $\delta^{15}N$ values in the soil-plant system between the sites should be taken into account. First, there is the N mobility before the foliar senescence, being its concentration thus distributed: leaves > litter fractions > soil in all the three sites. Second, the differences in soil textures, where more clayey soil was found in the secondary forest than in the eucalyptus removal sites. Clayey soils have greater retaining capacity when compared with sandy ones, especially the N ammoniacal form (*Trumbore & Camargo, 2009*). Clay soils present more intense processes of mineralization, nitrification, and denitrification than do sandy soils in the same region, and that influences the forest productivity and nitrogen availability (*Keller et al., 2005*). This suggests that N and $\delta^{15}N$ are relatively higher in sites with greater clay concentration (*Martinelli et al., 1999*; *Davidson et al., 2007*; *Houlton et al., 2007*; *Craine et al., 2009*). Nonetheless, greater water availability also tends to increase N concentration and $\delta^{15}N$ values in leaves and soils (*Austin & Vitousek, 1998*; *Handley et al., 1999*; *Amundson et al., 2003*; *Houlton et al., 2007*; *Nardoto et al., 2008*).

Even though the relationship between clay and soil $\delta^{15}N$ might be promote higher N fractionation (via nitrification, denitrification, and volatilization), in clay-rich soils the relative proportion of N fractionation might not be directly influenced by clays, but by the stability of clay particles containing $^{15}N$-enriched organic matter. As a result of the higher organic matter decomposition and/or higher clay concentrations, it is possible for soils from hotter and/or drier ecosystems to have a greater proportion of N in the organic matter associated with minerals, as opposed to having a higher proportion of N that is lost by fractionation processes (*Craine et al., 2015*).

The lower C/N litter ratio limits the N in the soil, thus raising the organic matter decomposition rates by the biota (*Berhe, 2012*). The C/N ratio in the leaves was greater in the M12 site. Higher C/N ratio values in wood fractions were obtained in the eucalyptus removal sites, in comparison with those of the secondary forest. That demonstrates the slow eucalyptus branch wood decomposition compared to that of the native species. Therefore, the secondary forest site, with less C/N ratio in the litter, has its N-rich organic compounds consumed rapidly; while the excess of N is mineralized, increasing hence the $\delta^{15}N$ value in the soil. That explains the negative relationship between $\delta^{15}N$ and the C/N ratio, which is also reported by big-scale evaluations (*Amundson et al., 2003*; *Craine et al., 2015*).

Whenever both soil and plant contain high $\delta^{15}N$ values, one may infer the presence of organic matter and decomposition activity in the soil, aside from variables such as temperature and gas flow into the atmosphere, which also interfere in the system (*Craine et al., 2015*). More humid soils (clayey) usually have more nitrogen gases emissions than those sandy soils (*Davidson et al., 2000*; *Keller et al., 2005*). Soils with larger quantities of silt and clay present a lower C/N ratio (*Stemmer, Gerzabek & Kandeler, 1998*; *Christensen, 2001*), as observed in the study herein.

The [15]N-rich organic matter decomposition, in general, has smaller $\delta^{15}N$ values in mountainous ecosystems from tropical forests than do a wide range of tropical forests in low lands worldwide with average values ranging from 3‰ to 14‰ (*Martinelli et al., 1999*; *Nardoto et al., 2014*). Such low $\delta^{15}N$ values were reported in hilly ecosystems and were related to both high concentration of $NO_3^-$ in soil solution and significant N lixiviation rates (10–15 kg N ha$^{-1}$ year$^{-1}$) (*Gütlein et al., 2016*).

A significant increase in soil $\delta^{15}N$ has also been observed in forests where recent fire events took place (*Hemp, 2005*; *Zech et al., 2011*). Vegetation burning may cause loss of gaseous $NO_2$ resulting in depleted values of $\delta^{15}N$ (*Zech et al., 2011*). Some disturbances in the ecosystem (forest fire, selective cutting, and others) trigger changes in the vegetal coat which affects the organic matter dynamics, and which may be regarded as variations in the C/N ratio in the soil (*Saiz et al., 2016*). Hence, this work suggests that the high $\delta^{15}N$ values in leaves and soil from the secondary forest are a consequence of relatively clay-rich soil combined with a moderate water deficit (*Ometto et al., 2006*; *Nardoto et al., 2008*; *Quesada et al., 2012*) which, consequently, would then lead to greater availability and loss of N (*Ometto et al., 2006*; *Nardoto et al., 2008*).

## The alteration in vegetation after eucalyptus removal modified the soil properties and furthered the decomposition processes of soil organic matter

The higher concentration of C and N in the soils of the M12 site is associated with two factors mainly: (1) greater eucalyptus litter entry in the system regarding eucalyptus logging, once only eucalyptus tree trunks were removed from the site in order to be commercialized; and (2) longer period since eucalyptus removal in this site, in comparison with the M3 site, which enabled the decomposition and incorporation of the M12 site litter-origin nutrients by the soil. However, the soil C concentration (3.3%) in the M12 site is among the lowest values reported for tropical forest soils, which vary from 3.0% (*Mazurec, 2003*) to 9.0% (*Clevelario Júnior, 1996*). A similar concentration of elemental N in the soil, found in the M12 site (0.3%) had been reported by *Villela et al. (2001)* in superficial soils of eucalyptus crops (0.31%) in the same studied site (Rebio União). The concentration of elemental N is greater in the leaves than it is in the soil and the litter fractions, which may be explained by the translocation of this mobile element in the plant before the foliar senescence (*Aerts & Chapin, 2000*; *Satti et al., 2003*; *Kuang et al., 2010*). The great necessity of N by this organ is due to the photosynthetic process in which many enzymes are entailed, as protein molecules with a catalytic function that contain N in their structures (*Paul & Pellny, 2003*).

The biotic and abiotic conditions fostered the decomposition of [13]C-enriched compounds in the M3 site. In more sandy-textured soils (M3 site), the organic matter tends to decompose more rapidly, which has less capacity to retain organic matter (*Rosell, Galantini & Iglesias, 1996*). The more intense presence of sand also favours the lixiviation of more soluble organic compounds, which might dissipate after the litter deposition. Moreover, the greater presence of grassy ($C_4$ plants) in the M3 site soil, compared with that of the M12 site, presented a higher value for $\delta^{13}C$ in the soil because

the $C_4$ species showed less isotopic discrimination than did $C_3$ species. From that, the M3 site soil had a higher $\delta^{13}C$ value (*Farquhar, 1983*). Studies on the Atlantic forest soils (*Vitorello et al., 1989*; *Bonde, Christensen & Cerri, 1992*) and the Amazon forest soils (*Desjardins et al., 1996*) stated that soils with greater clay quantity had higher values for $\delta^{13}C$. Even so, it was not confirmed by this research, where the $\delta^{13}C$ value was higher in the M3 site soil, which had a more considerable amount of sand. Certainly, the presence of both grassy species and sandy soil in the M3 site was an influencing factor upon $\delta^{13}C$ values in this soil.

The relationship between the C/N ratio and the $\delta^{15}N$ may be used as a tool for verifying degradation and stabilization of the soil organic matter in disturbed systems, in comparison with the secondary forest, which has already been observed (*Conen et al., 2008*). In the present study, the soil $\delta^{15}N$ increased along with the decrease in the soil C/N ratio soil. *Craine et al. (2015)*, studying 6,000 soil samples from 910 sites, had also noted the increase in soil $\delta^{15}N$ associated with a decrease in the soil C/N ratio, and as a consequence of greater stabilization of the soil organic matter. Soils with high clay content, as it is the case of the secondary forest, also have high soil $\delta^{15}N$. The clay and silt fractions are frequently associated with this soil organic matter, thus hampering the decomposition process (*Liao, Boutton & Jastrow, 2006*).

The organic matter with a longer incorporation time into the soil should be a better indicator of the isotopic signature of the decomposer than of the initial input from the fall of branches and plant leaves (*Craine et al., 2015*). For this reason, the soil organic matter fractions generally show higher $\delta^{15}N$ values in humidified soils (*Liao, Boutton & Jastrow, 2006*; *Marin-Spiotta et al., 2009*). *Liao, Boutton & Jastrow (2006)* quantified C and N stable isotopes in the soil, as well as their granulometry, in sites where tree species and bushes ($C_3$ plants) were replaced by pastures ($C_4$ plants). According to these authors, the soils with higher silt and clay fractions were those with higher $\delta^{15}N$ values, thus suggesting organic matter stabilization (*Marin-Spiotta et al., 2009*).

The data presented herein indicate the alteration of C and N dynamics in the ecosystem considering land use (presence and subsequent removal of the exotic eucalyptus species from the Atlantic forest fragment). It is important to emphasize the need for caution in the interpretation of these data since there are limitations around this type of experimental design. In studies that make use of real situations in the field, true replicates (sites) are difficult to work. In addition, this study addressed isotopic and soil granulometry analysis. More robust information can be obtained when other indicators and characteristics of vegetation and soil are also analyzed, including functionality attributes, photosynthetic performance, details of soil organic matter composition, among others. However, it is of utmost importance to understand how this exotic species affects the functioning of the ecosystem since eucalyptus has been planted all over the world for commercial or forest restoration purposes, either as monoculture or in symbiotic consortia with native tropical forests (*Guariguata, Rheingans & Montagnini, 1995*; *Parrotta, Turnbull & Jones, 1997*; *Feyera, Beck & Lüttge, 2002*; *Baptista & Rudel, 2006*). The substitution of native forest cover by the exotic eucalyptus species has been widely used in order to give financial returns to the landowners (*Baptista & Rudel, 2006*). In some cases, the

economic and environmental purposes are enhanced simultaneously, being the eucalyptus used for its rapid growth in deforested areas (*Feyera, Beck & Lüttge, 2002*), facilitating later forest restoration, as seen in plantations of this exotic species in Central America and the Caribbean (*Guariguata, Rheingans & Montagnini, 1995*; *Parrotta, Turnbull & Jones, 1997*). The fact that tropical forests are being the focus of recovery efforts after disturbances and degradation is mainly due to the urgent need to mitigate the consequences of climate change. In this sense, the data presented here are important to improve the understanding of the beneficial effect of the eucalyptus removal in the ecosystem.

## CONCLUSIONS

The removal of the exotic species altered temporally and differently the isotopic and elemental ratios of C and N in the different ecosystem compartments due to the floristic composition (appearance and abundant presence of grassy in the M3 site), the alteration in the environmental conditions (irradiance, water availability, and temperature), variations in mineralization and nitrification rates, and of the decomposition and input from the organic litter.

The soil $\delta^{13}C$ was higher in the site where the eucalyptus was harvested due to the facilitation of the appearance of grasses with photosynthetic $C_4$ syndrome that were decomposed and incorporated in the soil. Under the environmental conditions studied, such decomposition and incorporation occurred in a short period of time, as the highest values of $\delta^{13}C$ were observed in the area where the eucalyptus was removed 3 months ago (M3 site), but no later than 12 months (M12 site). Another aspect that contributed to the increase of soil $\delta^{13}C$ where the eucalyptus was harvested was the higher irradiance input, which increased the values of $\delta^{13}C$ in the photosynthesizing leaves that will be decomposed afterward.

The soil $\delta^{15}N$ also increased according to the increase in the time of eucalyptus harvesting (higher in the M12 site than in the M3 site), altering the dynamics of the organic matter in this compartment.

Therefore, the dual-isotope approach was able to generate a more integrated picture of the impact in the ecosystem after the eucalyptus removal in this secondary Atlantic forest. Finally, this tool should be considered as an option for future eucalyptus removal studies, taking into account the particularity of the data generated given the absence of sample replicas. Nevertheless, our data were able to detect the influence of this exotic species on the isotopic changes of C and N in all compartments of the ecosystem.

## ACKNOWLEDGEMENTS

The authors thank Dr. José Luiz Alves da Silva for preparing Fig. 1.

### Funding

This work was supported by the Laboratory of Environmental Sciences at the Graduate Programme in Ecology and Natural Resources of the Universidade Estadual do Norte

Fluminense Darcy Ribeiro, and the União Biological Reserve. Carlos E. Rezende received financial support from CNPq (305217/2017-8) and FAPERJ (E-26/202.916/2017). Angela P. Vitória was supported by CNPq (301169/2016-0). This study was financed by the Brazilian Federal Agency for Support and Assessment of Post-graduate Education Brazil (CAPES)-Finance Code 001. The funders had no role in study design, data collection and analysis, decision to publish, or preparation of the manuscript.

## Grant Disclosures

The following grant information was disclosed by the authors:
Laboratory of Environmental Sciences.
Natural Resources of the Universidade Estadual do Norte Fluminense Darcy Ribeiro.
União Biological Reserve.
CNPq: 305217/2017-8.
FAPERJ: E-26/202.916/2017.
CNPq: 301169/2016-0.
Brazilian Federal Agency for Support and Assessment of Post-graduate Education Brazil (CAPES): 001.

## Competing Interests

Carlos Eduardo de Rezende and Gabriela Bielefeld Nardoto are Academic Editors for PeerJ.

## Author Contributions

- Milena Carvalho Teixeira conceived and designed the experiments, performed the experiments, analyzed the data, prepared figures and/or tables, and approved the final draft.
- Angela Pierre Vitória conceived and designed the experiments, analyzed the data, prepared figures and/or tables, authored or reviewed drafts of the paper, and approved the final draft.
- Carlos Eduardo de Rezende conceived and designed the experiments, authored or reviewed drafts of the paper, and approved the final draft.
- Marcelo Gomes de Almeida analyzed the data, authored or reviewed drafts of the paper, and approved the final draft.
- Gabriela B Nardoto analyzed the data, authored or reviewed drafts of the paper, and approved the final draft.

## Field Study Permissions

The following information was supplied relating to field study approvals (i.e., approving body and any reference numbers):

Field experiments were approved by SISBIO/ICMBio/MMA to develop the project "Capacidade aclimatativa de espécies da Mata Atlântica após manejo de eucalipto na Reserva Biológica da União (REBIO)" at the RESERVA BIOLÓGICA UNIÃO, Rio de Janeiro, Brazil (field study approval number 37996-3).

## Data Availability

The raw data is available in the Supplemental Files.

## Supplemental Information

Supplemental information for this article can be found online at http://dx.doi.org/10.7717/peerj.9222#supplemental-information.

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
