# Peer review of "Consequences of removal of exotic species (eucalyptus) on carbon and nitrogen cycles in the soil-plant system in a secondary tropical Atlantic forest in Brazil with a dual-isotope approach"

_PeerJ, doi:10.7717/peerj.9222_

## Round 0.1 · original submission · Major Revisions

Dear Dr Nardoto,

Your manuscript has been reviewed by one external reviewer who found that your work has merit but also suffers from some deficiencies. As the reviewer has provided a thorough and extensive review, I have decided to take my decision based on this single review plus my own evaluation of your manuscript.
Upon careful consideration, I am ready to consider to evaluate a revised version that would take into account the comments and suggestions made. In particular, please carefully address the following:

Major comments from the external reviewer that I also want to stress:
* The English language should be improved (there is a lot of grammatical syntax issues). Please make sure that the article is read and corrected by an English speaker (and as usual, please provide the name of the person/organisation mobilized).
* In figure and table captions, please check the number of individuals and replicates per site.
* For live leaves, 5-7 leaves per individual were sampled. Please better justify to what extent it is representative, and detail which type of species have to be sampled for live leaves.
* You should be more critical about the results obtained and replicates (just one date for sampling).
* Please consider including a map with the localization of sites.

* I just think that you can disregard the reviewer's suggestion to cite figures or tables in the discussion; just make clear on which results each discussion part is based.

More generally, the reviewer made many constructive suggestions and provided an attached file with lot of comments, suggestions questions and corrections that will help you improving the ms.

In addition from my side:
* Lines 28-29: "the effects from forest management (removal) of exotic species" ==> be straightforward "the effects of the removal of the exotic species...". The term "forest management" is too general, so please use "eucalyptus removal" whenever this is what you refer to.
Lines 34-35 (and 146-147): "a secondary forest in the Atlantic forest, and two managed sites in different times: 12 and 3 months (M12 and M3, respectively)" ==> Do you mean : "a secondary forest in the Atlantic forest, and two sites where eucalyptus had been removed 12 and 3 months ago (sites M12 and M3, respectively)" ?
* A major comment is that you used only one site per treatment (i.e. one secondary forest site, one M12 site and M3 site), right? Thus, replicates are actually pseudo-replicates. Although this type of design is often the case in ecological studies, I would like you to (1) discuss to what extent no environmental gradients are expected to occur across sites (for instance do you have initial soil characteristics BEFORE eucalyptus removal?), and (2) recognize in the discussion the limitation of this type of experimental design and why true replicates (sites) are difficult to work with for this type of study. Providing a map of the sites with distances between sites is here compulsory. In addition, the use of only one sampling date is a limitation, e.g. for leaf isotopic signals, which should be discussed.
*Lines 50-51 and discussion: "after removing exotic plants in secondary Atlantic forests" ==> do you think that your results can be generalized to all types of exotic tree species? Cannot this strongly depend on, e.g., the quality of the foliage material let on the ground (i.e. exotic tree species which N-rich versus N-poor-foliage)? Please reinforce the discussion on this aspect.
* Lines 185-186: Like the external reviewer, I would like you to detail why leaf samples were collected at the third sun-exposed photosynthetically active pair of leaves and provide references. Refer to studies showing the great variation that exists for D13C within single tree crown due to variations in environmental conditions (doi.org/10.1046/j.0016-8025.2001.00756.x and https://link.springer.com/chapter/10.1007/978-94-007-1242-3_10) to explain why 'normalizing' these conditions is fundamental for this type of study.
* Section 'Data analysis': please detail the testing of data normality, required before ANOVA.
* Lines 246, 252-253 and elsewhere: please refrain from discussing your results in the result section ; insert these aspects to reinforce the discussion section.
* Lines 288-291: please based these assertions on your results. To some extent, your data allow unravelling sources of changes in D13C signals. For instance, you can explain that if the main sources of variations occur decomposition, this would mean similar D13C signals of living material (like leaves) but diverging signals of decomposed material. On the reverse, importance of irradiance, humidity and tree ecophysiology would lead to altered isotopic signals in living material (which could be tested in particular if using some same tree species between sites). More generally, please expand the discussion to reinforce the rationale supporting your conclusions.
* Lines 333-334: " The leaves present higher lignin concentrations than stems and roots, which possess more cellulose" ==> sounds odd?
* Lines 332-343: sounds more like a course on 13C signal variation between tree organs than a discussion focused on your objective which is effects of exotic species removal.
* Line 470: "The temporal variation in forest management": rather, "The removal of the exotic species". Also, as suggested by the external reviewer, you could reinforce a bit the conclusions by adding possible implications for N and C cycling that just concluding that there is and "impact from the forest management in the soil’s carbon and nitrogen fractioning processes, which alters the dynamics of the soil’s organic matter." this is too general and not unexpected; please be a bit more specific in the conclusions and raise possible implications.
* In the figures: you seem to draw a single relationship each time; however, all the points (e.g., compartments in Fig 1) do not seem to be well fitted by a single line; modify accordingly.
* Moreover, you should particularly test to what extent exotic species removal/site affects each relationship; this is done in Fig 2 for 15N versus C:N ratio, but I miss a test for the relationships presented in Figure 1 ==> please directly test the significance of site/management on each relationship presented in Fig 1, provide results of the statistical tests, and present how the relationships differ between treatments when relevant. This would be more consistent with the main objective of your ms.

I look forward to receiving a revised version of your manuscript along with a point by point response to all the comments and suggestions made. Please provide a word document of your revised ms.

best regards,

Xavier LE ROUX


Reviewer 1 ·

Basic reporting

The English should be improved because there is a lot of grammatical syntax issues in the text. Notably, authors use the possessive form (for example “elements’ dynamics”, “ecosystem’s compartments”, …), but this form is only used with nouns referring to people, groups of people, countries, and animals, and it is not appropriate in the context of the article. There are some English mistakes: ex: ration for ratio, grater for greater, ... For a better reading of the article, it should be that the article is read and corrected by an English speaker.
Moreover, the article has to be read correctly by authors because there are some repetitions (for example L20 to L24 repetition with L25 to L30).
Authors use lot of acronyms that have to be defined they were used. For example, page 5 – line 175: DBH, page 5 – line 178: IVI, ...

The introduction of the article is good and the subject and aims are correctly brought. In the introduction, literature references are correctly used to provide the field background and the context.

The structure of the article is in format of “standard sections” given in the instructions for authors.
Figures are relevant to the content of the article and have a good resolution. However, they would be clearer if the legends were included inside. In title, authors have to check the number of replicates and precise if it is per site, the number of individuals per site, … Same in titles of tables, check and precise at what correspond the number of replicates.
File with raw data is correctly presented and seems to be complete. But it would be interesting that authors precise at what type of species correspond the live leaves in the first sheet of the excel file.

Results are relevant and defenced correctly hypothesis. It will be good to cite figures or tables in the discussion, to see on which results the discussion is based.

Experimental design

Isotopic composition method used to study dynamics of carbon and nitrogen in soils is well known and not original but the using of this in different compartments of soil-plant systems (different parts of litter, live leaves, soils) is very interesting and less realized. Moreover, generally authors are more focused on one element (carbon or nitrogen) but it is not very common to have a comparison between the dynamics of these two elements. Past management with removing of native forest replaced by species for commercial is an important issue in tropical region. And now, there is an effort to restore the native forest in removing the commercial species. This is the case in the Uniao Biological Reserve, Brazil, since 2015 with the removing of eucalyptus species. But the impact of this management on carbon and nitrogen dynamics is poorly documented and studied because very recent. So, this article is interesting and important for the domain of study of the global cycles of nutrients and impact of anthropic management (disturbance, restauration) on them.
But it will be interesting that authors have a more global conclusion and include the impact of this management on the global carbon and nitrogen cycles and global changing.

The investigation was done on a large number of samples for soils and for litters (25 replicates per site). So, the investigation seems to be conducted rigorously for this part. But for live leaves, only 5 individuals per species and 5 to 7 leaves per individual were sampled. Authors have to justify if it is representative. Moreover authors have to precise which type of species have to be sampled for live leaves (in Materials and Methods part and in the excel file for raw data).

“Methods” part has to be well described to have sufficient information and details to be reproductible. Authors have to justify the methods used.
Authors have to precise which soil classification they used to determine their soils (L162).
For part 2 “Determination of elemental and isotopic composition …”, authors have to be more consistent and justify the methods they used.
- Why leaf samples were collected at the third sun-exposed photosynthetically active pair of leaves? References? Why 5 individuals and 5 to 7 leaves? Is it to have the same weight? Is it representative of sites?
- 25 samples of litter or soil, is it representative? Justify.
- Methods used to dry samples have to be justified notably with references.
- Why soil have to be dried at 80°C and no at 105°C (classic method to remove all waters contained in soils and notably pellicular water)? Soil samples have been decarbonated? If no, authors have to justify why (because C analyses – concentration and isotopic composition – are supposed to be done on the organic fraction but without decarbonation the inorganic carbon is taken into account).
- Why is there a maceration step for leaf and litter samples? Explain. The maceration was done in what: water, other liquid?
- Add values of concentration and/or isotopic compositions of standards: PDB, atmospheric N, Wheat Flour, Low Organic Content Soil.
For part 3 “Determination of the soil granulometric fractions”, authors have to be more precise.
- What are certified samples: JIS 11, Locopdium, glassbeads? Precise size of their grains?
- No JISS but JIS for Japanese Industrial Standard.
- Why authors do not use all standards from JIS?
In part 4 “Data analysis”, authors have to change the title, because it is not appropriate for this part. Example: “Statistical analysis”.
- Why are the “compartments, study sites and granulometric fractions” used for factors? Explain.

Validity of the findings

Results are clearly exposed and support the discussion and the conclusions.
The data on which the discussion and the conclusions are based are in the correct form for the scientist community interested in nutrient cycles, soils, soil-plant systems and impact of disturbances on them. The data are robust and controlled statistically via ANOVA (two ways), followed by Tuckey test. Authors have to justify why they use these factors (compartments, study sites and granulometric fractions) to statistical analysis.
The conclusion is very short. It will be interesting that authors extend their conclusion in the more general context: global nutrient cycles, global changing, ...

Additional comments

The manuscript of Milena Carvalho Teixeia et al., examines the impact of management (removal) of exotic species in the soil-plant system in a secondary tropical Atlantic forest. The study interests in a dual-isotope approach (d13C and d15N) to reveal the consequences of this management on C and N cycles. This kind of study is of interest to better understand the effects of anthropic disturbance or restauration on natural forests (especially in tropical environment) and C and N cycles in these contexts.
The title should be rewritten (to replace the “sentence” title), as for example: “Consequences of management of exotic species on carbon and nitrogen cycles in the soil-plant system in a secondary tropical Atlantic forest (ReBio Uniao, Brazil), with a dual-isotope approach”.
Maybe, it will be interesting to have a map with the localization of sites.
The discussion is interesting and has a good comparison with literature, but should be more consistent and long. Indeed, the authors should be more critical about their results and replicates (just one date (December 2013) for sampling).
The English language should be improved because there is a lot of grammatical syntax errors.

Detailed comments:
Detailed comments and corrections are added in the attached file. When comments are putted in quotations marks, it is a proposition to rewrite the text or a correction of the text. When there are no “quotation marks”, it is a question, general comment, …
It will be good that authors do not change the name if the site. Example: choose “second growth forest” or “secondary forest”, but do not use the two expressions.
In English, it is better when sites are cited to use “the M3 site”, “the M12 soil”, … Check the text.
There are some mistakes in figures and they could be more consistent:
- Add legends in the figures.
- Titles have to be checked, notably the number of replicates. Precise if it is per site, the number of individuals per site or per individuals, … Same in titles of tables, check and precise at what correspond the number of replicates.
- Figure 1F: N (%) vs d15N (‰) for Site M3: closed triangles (soil) seem to have the same nitrogen isotopic composition = 0 ‰, but it is not the case on the Figure 1E where the values seem vary between 0 to 1.9 ‰. There is a problem.
- Figure 3: clay < 4 µm. Why not < 2 µm as in literature?

Titles of tables have to be checked and more relevant.
Table 1: irradiance values were obtained on only one sunny day. Is it representative? Justify. There is no a meteorological station close to the studied sites to have a real mean value on few years?
Why is there no value for the surface area of the secondary forest site?

Annotated reviews are not available for download in order to protect the identity of reviewers who chose to remain anonymous.

---

## Round 0.2 · Minor Revisions

Dear authors,
Your manuscript has been much improved, which is recognized by the reviewer who has re-evaluated it. However, the reviewer also raised a series of remaining issues which still have to be addressed:

* Overall, the English language has been improved, but there are still mistakes that should be corrected. Before accepting the article, this should be carefully corrected.

* Also please correct the (too many) errors in figures, figure and table captions, acronyms, etc.

I hope you appreciate that the objective of this additional iteration on your manuscript is to further improve it, and I look forward to receiving the revised version.

Best regards
Xavier LE ROUX

Reviewer 1 ·

Basic reporting

Authors have done a good revision of their English in comparison of their first version, but there still are some mistakes/errors in text:
- Check the writing of the carbon isotopic composition, because the size of 13 (δ¹³C) is smaller than the 15 for the nitrogen isotopic composition.
- Precise “in the M3 site” (L.31, L.299, L.328, L.338, …), “in the M12 site” (L.35, L.273, …)
- Check if it is better to use the term “cycle” or “recycling” rather than “cycling”
Examples: L.32; L. 80: “The C and N cycles”; L.88: “the soil texture influences on the recycling or cycles” …
- Check if it is better to use the term “Fractionation” rather than “fractioning”
Examples: L. 108 “thus resulting in N fractionation” no fractioning
- Check if it is better to use the term “decomposition” rather than “decomposing”. Example: L. 120 “high decomposition rates”
- L. 119-120 “tend to provide favourable conditions for decomposition activity by soil organisms”
- L. 85-86: “the litter decomposed material” no “decomposing”
- L. 130-132: “In the União Biological Reserve (Rebio União), a fragment of the Atlantic forest with integral nature protection located in southwest Brazil, presents a specific forestry management which consist on the removal of eucalyptus trees (Corymbia citriodora) until 2015 aiming to restore the native flora.”
- L. 159-164: sentence too long separate in 2 sentences. “We realized this research in three latosolic dystrophic red-yellow argisoil sites, classified according to Miranda et al. (2007) and Embrapa (1999). One site is a well-preserved secondary forest site; whereas the other two are eucalyptus-harvested sites, by means of clear-cutting, with a single difference related to after-handling time: the M12 (where eucalyptus had been removed 12 months before our evaluation); and the M3 (where eucalyptus removal had happened 3 months before assessment) (Fig. 1, Table 1, Supplementary Material 1).”
- L. 229 to 232: caution values are written with coma. Example: 39,38 ± 0,19 %  Change in 39.38 ± 0.19 %
- L.439: NO3- the 3 is in subscript and the minus in exponent
- L.464: “The concentration of elemental N is greater in the leaves than it is in the soil and the litter fractions …”
- L.489: 6,000 in English for thousand numbers, it is a coma used not a point

Experimental design

Authors made it clear why they used this or that method.
- But, L.159, authors precised from which publication the classification of their soil comes from, but they have to precise the soil classification used: Brazilian Soil Classification System (dos Santos et al., 2018)? US Soil Taxonomy???
- L.212: 75 or 25 soil samples??
- In the text, it appears different terms for the supplementary materials: Supplementary Material 1 or 2, and DataS1, S2, ... Authors have to check which term is the good and do correction.

Validity of the findings

Authors have really improve their discussion and conclusion.

Additional comments

Authors have really improved their mansucript compare to the first version, and reply to reviewer and editor with good aguments.
But there are still some mistakes, notably for English (cf. 1. basic reporting) and Figures:
- There is a problem with figures: it was not possible to open Figures 2 to 5 (.EPS). Authors have to check the format for the submission.
- Figure 2: on the figure, the r value is wrong; it is not 0.03!!! The value has to be negative and maybe around -0.5 or -0.6
- Figure 3: in the title precise the definition of SE; on the figure, the ordinate is not the soil texture but the percentage of sand, silt and clay.
- Table 1: Acronym of site clear-cut 3 months ago is M3 and no 3M

---

## Round 0.3 · accepted · Accept

Dear authors,

I am pleased to inform you that, following the revision made based on the reviewer’s comments, your manuscript is now acceptable for publication in PeerJ.

Best regards

Xavier LE ROUX